# Transcriptomic Profiling Uncovers Molecular Basis for Sugar and Acid Metabolism in Two Pomegranate (*Punica granatum*) Varieties

**DOI:** 10.3390/foods14101755

**Published:** 2025-05-15

**Authors:** Ding Ke, Yilong Zhang, Yingfen Teng, Xueqing Zhao

**Affiliations:** 1Co-Innovation Center for Sustainable Forestry in Southern China, Nanjing Forestry University, Nanjing 210037, China; keding@njfu.edu.cn (D.K.); zyl23@njfu.edu.cn (Y.Z.); tengyingfen@njfu.edu.cn (Y.T.); 2College of Forestry, Nanjing Forestry University, Nanjing 210037, China

**Keywords:** pomegranate, soluble sugars, organic acids, transcriptomic profile, differentially expressed genes, transcription factor

## Abstract

Soluble sugars and organic acids constitute the primary flavor determinants in fruits and elucidating their metabolic mechanisms provides crucial theoretical foundations for fruit breeding practices and food industry development. Through integrated physiological and transcriptomic analysis of pomegranate varieties ‘Sharp Velvet’ with high acid content and ‘Azadi’ with low acid content, this study demonstrated that the differences in flavor between the two varieties were mainly caused by differences in citric acid content rather than in soluble sugar content. Transcriptome profiling identified 11 candidate genes involved in sugar and acid metabolism, including three genes associated with soluble sugar metabolism (*FBA1*, *SS*, and *SWEET16*) and eight genes linked to organic acid metabolism (*ADH1*, *GABP1*, *GABP2*, *GABP3*, *GABP4*, *ICL*, *ME1*, and *PDC4*). These data indicated that differences in citric acid content between the two varieties mainly stemmed from differences in the regulation of the citric acid degradation pathway, which relies mainly on the γ-aminobutyric acid (GABA) branch rather than the isocitric acid lyase (ICL) pathway. Citric acid accumulation in pomegranate fruit was driven by metabolic fluxes rather than vesicular storage capacity. Weighted gene co-expression network analysis (WGCNA) uncovered a significant citric acid content associated module (r = −0.72) and predicted six core transcriptional regulators (*bHLH42*, *ERF4*, *ERF062*, *WRKY6*, *WRKY23*, and *WRKY28*) within this network. Notably, *bHLH42*, *ERF4*, and *WRKY28* showed significant positive correlations with citric acid content, whereas *ERF062*, *WRKY6*, and *WRKY23* demonstrated significant negative correlations. Our findings provide comprehensive insights into the genetic architecture governing soluble sugars and organic acids homeostasis in pomegranate, offering both a novel mechanistic understanding of fruit acidity regulation and valuable molecular targets for precision breeding of fruit quality traits.

## 1. Introduction

Pomegranate (*Punica granatum* L.), a deciduous shrub or small tree belonging to the Lythraceae family (genus *Punica*), is a horticulturally important crop originating from Central Asia that is subsequently domesticated in Mediterranean and Middle Eastern regions [1]. As one of the earliest cultivated fruit species in human history [2], pomegranate has gained global agricultural significance due to its multifunctional value, encompassing economic, nutritional, medicinal, and ornamental applications [3]. The fruit is particularly renowned for its high anthocyanin content [4], which confers potent antioxidant properties [5] and demonstrates therapeutic potential in mitigating inflammation and preventing carcinogenesis [6,7]. The global pomegranate market has witnessed sustained growth, fueled by increasing consumer demand for functional foods. Given its commercial importance, substantial research efforts have been devoted to elucidating its bioactive compounds and optimizing cultivation practices.

Soluble sugars and organic acids are the primary determinants of fruit flavor, playing pivotal roles not only in regulating the balance between sweetness and acidity but also in influencing texture, coloration [8,9], processing characteristics, marketability, and economic value. Fruit flavor profiles are primarily shaped by complex interactions involving the ratio of soluble sugars to organic acids, their specific molecular composition, and synergistic effects [10,11]. The predominant soluble sugars in fruit tissues include sucrose, glucose, and fructose [12,13]. In most fruit crops, these metabolites serve dual physiological roles—acting both as energy substrates for respiration and as osmoregulators critical for maintaining cellular homeostasis during fruit development and postharvest storage. Sucrose is the dominant photoassimilate transported through the phloem [14], where it undergoes cytosolic metabolism before being sequestered in the vacuole [15]. Among organic acids, malic acid and citric acid are the most abundant [16,17], functioning as both respiratory substrates [18] and pH regulators [19].

The metabolic regulation of soluble sugars and organic acid accumulation in fruits involves a sophisticated, multilayered control network governed by coordinated processes including biosynthesis, catabolism, transmembrane transport, vacuolar sequestration, transcriptional regulation [20,21,22], and environmental modulation [23,24,25]. Currently, the development of various molecular biology tools including transcriptomics has greatly helped us to gain a deeper understanding of the regulatory mechanisms of soluble sugars and organic acids metabolism. Transcriptomics, which analyzes the composition and dynamic changes of the complete set of transcripts under specific physiological states, reveals gene expression patterns and regulatory networks, playing a pivotal role in functional gene discovery, expression divergence analysis, and molecular mechanism inference. For instance, transcriptomic studies in loquat [26], litchi [27] and sweet cherry [28] have identified multiple gene families that regulate soluble sugar and organic acid biosynthesis through differential expression. Comparative analyses reveal substantial interspecific variation in the key genetic determinants of these metabolic pathways, suggesting that the molecular control of soluble sugars and organic acid balance is influenced by complex interactions among species-specific genetic backgrounds, variety characteristics, and agronomic management practices [27].

Current applications of transcriptomics in pomegranate research have predominantly focused on abiotic stress tolerance [28], anthocyanin biosynthesis [29], fruit cracking mechanisms [30], and floral pigmentation [31], leaving a significant knowledge gap regarding the molecular basis of flavor compound accumulation in this species. This gap impedes targeted breeding efforts to meet consumer preferences for sweeter, less acidic varieties. Therefore, this study employs a comparative transcriptomic approach to investigate the molecular mechanisms underlying soluble sugars and organic acids metabolism in pomegranate, utilizing two varieties—‘Sharp Velvet’ with high acid content and ‘Azadi’ with low acid content—as experimental models. Based on the established patterns of fruit soluble sugars and organic acid accumulation, transcriptomic profiling was conducted to identify differentially expressed genes (DEGs), followed by weighted gene co-expression network analysis (WGCNA) to systematically characterize the key transcription factors (TFs) involved in citric acid metabolic regulation. Our findings provide novel insights into the genetic architecture controlling flavor compound biosynthesis in non-model fruit crops, while establishing a conceptual framework for cultivation practices targeting flavor optimization through metabolic engineering and molecular breeding strategies for developing novel pomegranate varieties with enhanced organoleptic qualities.

## 2. Materials and Methods

### 2.1. Fruit Sampling

Fruits from two pomegranate varieties ‘Sharp Velvet’ with high acid content and ‘Azadi’ with low acid content, matured in early October and were collected from commercial orchards in Liuhe District (118.76 N, 32.44 E), Nanjing, China. ‘Sharp Velvet’ displayed deeper red exocarp and arils than ‘Azadi’. All sampled trees were grown under standardized cultivation conditions in the same orchard block. For each variety, three uniform trees were selected as biological replicates. Fruits were sampled at six developmental stages, including 60, 75, 90, 105, 120, and 135 days after flowering (DAF). At each time point, five representative fruits per tree were harvested from different canopy positions (east, south, west, north, and top), immediately placed on ice, and transported to the laboratory. After removing the calyx, each fruit was quartered longitudinally. Arils from five fruits per replicate were pooled, randomly divided into two portions, and either flash-frozen in liquid nitrogen for RNA extraction or juiced through four layers of sterile cheesecloth. All samples were stored at −80 °C until analysis.

### 2.2. Determination of the Content of Main Components of Soluble Sugars and Organic Acids

The quantification of citric acid and malic acid was performed using an improved HPLC method [32,33]. Briefly, 1 mL of pomegranate juice was mixed with 3 mL of 20 mmol/L Na_2_HPO_4_ solution (pH 7.0) and vortexed for 30 s. The mixture was centrifuged at 7500× *g* for 15 min at 4 °C, and the supernatant was collected. This extraction procedure was repeated twice with 2 mL of fresh buffer each time. The combined supernatants were adjusted to a final volume of 10 mL with extraction buffer. Prior to injection, samples were filtered through 0.22 μm nylon membrane syringe filters. Chromatographic separation was achieved using a Waters HPLC system equipped with a 2489 UV detector and a Phenomenex Gemini C18 column (5 μm, 4.6 × 150 mm) maintained at 40 °C. The mobile phase consisted of 0.1% phosphoric acid: methanol (98%:2%) delivered isocratically at 0.7 mL/min. The detection wavelength was set at 210 nm with an injection volume of 10 μL. For calibration, stock solutions (2 mg/mL) of certified reference standards were prepared in the mobile phase and serially diluted to four concentrations (0.1, 0.2, 0.4, 0.8 mg/mL). Each standard was injected in triplicate to establish retention times and calibration curves (Appendix A). Linear regression analysis of peak area versus concentration yielded correlation coefficients for both analytes (Appendix A).

Fresh aril tissue (100 mg) was deseeded and cryogenically ground in liquid nitrogen, then homogenized in 0.9 mL ice-cold phosphate buffer (0.1 mol/L, pH 7.4). Soluble sugar contents were determined using commercial kits from Nanjing Jiancheng Bioengineering Institute (Nanjing, China): Glucose Kit (A154-1-1, glucose oxidase method) for glucose, Sucrose Assay Kit (A099-1-1, ultraviolet colorimetric method) for sucrose, and Fructose Assay Kit (085-1-1, ultraviolet colorimetric method) for fructose.

### 2.3. RNA Extraction, Library Construction and Sequencing

Based on preliminary biochemical profiling, transcriptome sequencing was performed on fruit samples collected at three critical developmental stages (90, 120, and 135 DAF) that exhibited the most pronounced soluble sugars and organic acids compositional differences between the two pomegranate varieties. RNA extraction and cDNA library construction were completed by Guangzhou Genedenovo Biotechnology Co., Ltd. (Guangzhou, China) and the constructed libraries were subjected to transcriptome sequencing using the Illumina NovaSeq6000 (version 1.0).

### 2.4. Transcriptomic Data Analysis and DEGs Identification

The raw sequencing reads were aligned to the reference genome using HISAT2 (version 2.2.1) [34] with default parameters. Transcript assembly was performed using StringTie (version 2.1.5) [35]. Gene expression quantification was conducted via RSEM [36], implementing expectation-maximization algorithms to calculate transcript abundances normalized as TPM (Transcripts Per Kilobase Million) values, which account for both sequencing depth and transcript length variations. Based on the TPM data, DEGs were screened based on the criteria of FDR < 0.05 and |log_2_Fold Changes| > 1. GO and KEGG databases were used to conduct an enrichment analysis of biological functions and metabolic pathways for the differentially expressed genes. The online website (http://www.omicshare.com/, accessed on 16 November 2024) was applied to Principal Component Analysis (PCA) and heat maps analysis.

### 2.5. WGCNA

Through WGCNA of 1166 TFs identified from pomegranate fruit comparative transcriptomes, we systematically identified candidate regulators of citric acid metabolism. The analysis was performed using the WGCNA R package (version 4.2.2) with TPM-normalized expression data [37], employing a soft-thresholding power of 8 to construct scale-free co-expression networks. Module–trait relationships were established by correlating module eigengenes with citric acid content profiles. Following rigorous screening criteria (module membership MM > 0.80 and gene significance GS > 0.65) as previously established [38].

### 2.6. Real-Time Fluorescence Quantitative Analysis (qRT-PCR)

Total RNA was extracted from pomegranate samples using the SPARKeasy New Plant RNA Rapid Extraction Kit (AC0305, Shandong Sikejie Biotechnology, Jinan, China) with on-column DNase I digestion to eliminate genomic DNA contamination. RNA purity was assessed by NanoDrop 2000 spectrophotometry (version 1.6), yielding A260/A280 ratios of 1.9–2.1, indicating minimal protein and organic compound contamination. Subsequently, cDNA synthesis was performed using SPARKscript II All-in-one RT Super Mix (AG0305, Shandong Sikejie Biotechnology, Jinan, China) following the manufacturer’s protocols. Quantitative real-time PCR analysis was performed using SYBR Green chemistry with the following thermal cycling protocol: initial denaturation at 94 °C for 3 min, followed by 40 cycles of denaturation at 94 °C for 10 s, annealing at 60 °C for 20 s, and extension at 72 °C for 20 s. Each reaction included three biological and technical replicates. The primers for qRT-PCR (Appendix A) were designed through the online website (https://www.ncbi.nlm.nih.gov/tools/primer-blast, accessed on 11 December 2024). Relative gene expression was calculated using the 2^−ΔΔCT^ [39] method with *PgActin* as the internal reference gene.

### 2.7. Statistical Analysis

Statistical significance was determined by one-way analysis of variance (ANOVA) using SPSS 21.0 software (IBM Corporation, USA) [40]. Significant differences between the groups were determined using Duncan’s new multiple-range test. Pearson correlation coefficients were calculated to evaluate associations between variables, and correlation heatmaps were generated with OriginPro 2024 (OriginLab Corporation, Northampton, MA, USA) [29].

## 3. Results

### 3.1. Dynamic Changes of Principal Soluble Sugars and Organic Acids Components During Developmental Period

Fructose, glucose, and sucrose were quantitatively identified as the predominant soluble sugars in pomegranate fruits, all displaying coordinated accumulation patterns during fruit development (Figure 1 and Appendix A). At maturity, the sugar concentrations varied significantly between varieties: fructose levels reached 36.19 mg/mL in ‘Azadi’ compared to 39.04 mg/mL in ‘Sharp Velvet’, while sucrose concentrations were 29.08 and 32.53 mg/mL, and glucose levels were 25.11 and 29.25 mg/mL, respectively. Early developmental stages revealed notable interspecific variation in fructose accumulation, though these differences gradually diminished during later phases. Particularly striking was the significant divergence in sucrose levels between varieties at the ripening stage, clearly indicating distinct regulatory mechanisms in sucrose metabolism.

The organic acids composition of pomegranate fruit comprises citric acid, malic acid, oxalic acid, and succinic acid, with citric and malic acids being recognized as the predominant organic acid components in most fruit species. Both varieties exhibited an overall decreasing trend in citric acid and malic acid concentrations during fruit development (Figure 1). The citric acid content in ‘Azadi’ and ‘Sharp Velvet’ pomegranates reached their maximum values at 105 and 60 DAF, respectively, and there were remarkable differences in citric acid content throughout the whole developmental period. The content of malic acid in both pomegranate varieties reached their maximum values at 60 DAF, with the content being 12.22 and 7.72 mg/mL, respectively.

### 3.2. Identification of DEGs

The sequencing process produced 110.10 Gb of raw data, which were subsequently processed to yield 110.75 Gb of high-quality clean data. The resulting sequences exhibited excellent quality metrics, with Q30 scores surpassing 93.58% for all samples. Genome alignment demonstrated exceptional performance, with mapping rates between 97.55% and 98.13% against the pomegranate reference genome. Notably, uniquely mapped reads accounted for 94.52% to 95.38% of total alignments (Appendix A).

PCA analysis of transcriptome data from ‘Azadi’ and ‘Sharp Velvet’ pomegranate varieties at three developmental stages (90, 120, and 135 DAF) showed a clear separation of samples according to developmental timepoints (Figure 2A). The tight clustering of biological replicates for each developmental stage confirmed the high reproducibility of our experimental results.

Transcriptomic analysis identified a total of 8644 DEGs across developmental comparisons. Temporal comparisons in ‘Azadi’ revealed 1982 (A120 vs. A90), 2819 (A135 vs. A120), and 4915 (A135 vs. A90) DEGs, while ‘Sharp Velvet’ showed 2071 (S120 vs. S90), 3175 (S135 vs. S120), and 5544 (S135 vs. S90) DEGs (Figure 2C). Notably, early developmental transitions (90–120 DAF) exhibited comparable numbers of up and downregulated DEGs, suggesting relatively stable transcriptional regulation during initial fruit maturation. However, later stages displayed progressive increases in both the number of DEGs and downregulation bias, indicating extensive transcriptional reprogramming during ripening.

Comparative transcriptomic profiling between ‘Azadi’ and ‘Sharp Velvet’ pomegranate varieties revealed distinct patterns of DEG expression across developmental stages (Figure 2B). When using ‘Azadi’ as the reference, we identified 1580 DEGs at 90 DAF, including 895 upregulated and 685 downregulated genes. The lowest inter-variety differential expression was detected at 120 DAF, comprising 1522 DEGs (819 upregulated and 703 downregulated). In contrast, the highest number of DEGs (2164) was observed at 135 DAF, with 1125 upregulated and 1039 downregulated transcripts. These findings demonstrate that while variety-specific differences accumulate during fruit development, the fundamental regulatory architecture remains similar, with a consistent predominance of upregulated genes across all comparisons.

### 3.3. Functional Enrichment Analysis of DEGs

Functional annotation of DEGs across all nine comparison groups revealed significant enrichment in Gene Ontology (GO) categories (Figure 3A and Appendix A). Among the top 20 most significantly enriched GO terms, we observed distinct distribution across the three major GO domains: nine terms belonged to molecular function, six to cellular component, and five to biological process.

KEGG pathway enrichment analysis of DEGs across all nine comparison groups revealed significant overrepresentation in metabolic pathways (Figure 3B and Appendix A). The most prominently enriched metabolic pathways included galactose metabolism, starch and sucrose metabolism, amino acid metabolism, pyruvate metabolism, glyoxylate and dicarboxylate metabolism, citric acid cycle, glycolysis and gluconeogenesis, and amino acid and nucleotide sugar metabolism. DEGs in these pathways were involved in the regulation of sugars and acids metabolism in pomegranate.

### 3.4. Screening of Candidate Genes for Soluble Sugars and Organic Acids Metabolism

Through comprehensive analysis of enriched soluble sugars and organic acids metabolic pathways, we identified 11 candidate genes with putative roles in pomegranate fruit quality regulation (Appendix A), including *ADH1* (*LOC116212354*), *FBA1* (*LOC116208886*), four *GABP isoforms* (*LOC116187566*, *LOC116194670*, *LOC116202008*, and *LOC116202067*), *ICL* (*LOC116194077*), *ME1* (*LOC116197013*), *PDC4* (*LOC116209682*), *SS* (*LOC116197401*) and *SWEET16* (*LOC116193209*). GO and KEGG analyses showed predominant enrichment of these candidate genes in metabolic pathways (Appendix A). All individual gene expression patterns are displayed in Figure 4 and Figure 5. In ‘Azadi’, six genes (*ADH1*, *FBA1*, *ICL*, *PDC4*, *SS*, and *SWEET16*) showed progressive transcriptional downregulation from A90 to A135 (Appendix A). ‘Sharp Velvet’ displayed minimal differential expression, with only three genes (*ICL*, *ME1* and *SWEET16*) meeting DEG criteria (Appendix A). These exhibited divergent regulation patterns. Cross-variety comparisons revealed consistent overexpression of all four *GABP* isoforms in ‘Azadi’, suggesting variety-specific regulatory roles (Appendix A).

### 3.5. WGCNA Identified TFs Related to Citric Acid Metabolism

The accumulation of organic acids in pomegranate is regulated by an intricate network involving biosynthesis, catabolism, and transport processes, with additional control exerted by TFs from the bHLH, ERF, MYB, and WRKY families. While malic acid and soluble sugar levels showed no significant differences between varieties, citric acid exhibited marked differential accumulation, prompting focused analysis of citric acid-associated transcriptional regulators from the initial pool of 1166 identified TFs. WGCNA was performed with scale-free topology fit (Appendix A) and soft-thresholding power of 8, which partitioned TFs into 9 co-expression modules (Figure 6A) ranging in size from 294 (turquoise) to 33 (grey) members. Module–trait correlation analysis revealed the green module as most significantly associated with citric acid content, containing 14 bHLHs, 8 ERFs, 3 GATAs, 11 MYBs, 3 TCPs, 8 WRKYs and other TFs (Figure 6D). KEGG enrichment showed predominant representation in basal transcriptional processes (Appendix A). Employing stringent selection criteria from WGCNA with module membership (MM) > 0.8 and gene significance (GS) > 0.65 [38], integrated with differential expression profiling, we identified six core transcription factors demonstrating strong functional associations with citric acid metabolism (Figure 7, Appendix A). The candidate regulators comprise a bHLH family representative (*bHLH42*), two ERF family proteins (*ERF4* and *ERF062*) and three WRKY family members (*WRKY6*, *WRKY23*, and *WRKY28*), representing key transcriptional components potentially governing organic acid homeostasis in the studied system.

### 3.6. Correlation Analysis

Correlation analysis of soluble sugars, organic acids, and candidate gene expression levels revealed distinct regulatory patterns (Figure 8). The expression levels of four *GABP* isoforms exhibited significant negative correlations with citric acid content, whereas, the expression abundance of *ADH1* and *PDC4* showed no significant correlations. Notably, *ICL* expression level demonstrated a strong positive correlation with citric acid levels, while *ME1* was negatively correlated with both citric acid and malic acid content. *FBA1* and *SS* showed highly significant negative correlations with sucrose, fructose, and glucose content. The expression level of *SWEET16* was specifically correlated with fructose and glucose content.

Correlation analysis revealed significant regulatory associations between the identified TFs and citric acid accumulation (Figure 9). Specifically, the expression level of *bHLH42*, *ERF4* and *WRKY28* exhibited significant positive correlations with citric acid content, whereas *ERF062*, *WRKY6* and *WRKY23* demonstrated significant negative correlations. This bidirectional regulatory pattern suggested that citric acid metabolism was governed by a sophisticated and finely tuned transcriptional regulatory network.

### 3.7. qRT-PCR Validation of DEGs

To further validate the accuracy of the transcriptome data, we selected six candidate genes and TF (*ADH1*, *ERF062*, *ICL*, *ME1*, *PDC4*, and *SS*) associated with soluble sugars and organic acids metabolism in pomegranate fruits for qRT-PCR analysis (Figure 10). The qRT-PCR results were largely consistent with the transcriptome data, confirming the reliability of our transcriptomic findings.

## 4. Discussion

Metabolic profiling revealed dynamic changes in soluble sugars and organic acids during fruit development in both pomegranate varieties, with patterns similar to those reported in many fruits such as loquat [26] and apple [41]. The observed inverse relationship between progressively increasing soluble sugars and decreasing organic acids across developmental stages suggested potential metabolic conversion of organic acids to soluble sugars via gluconeogenesis [16,42,43]. Notably, while the two varieties show minimal divergence in soluble sugar profiles, they exhibit substantial differences in organic acid accumulation, indicating that variety-specific flavor variation is predominantly governed by organic acid metabolism [17,44,45]. In particular, this study identified citric acid as the pivotal flavor determinant driving acidity divergence between varieties. Previously, Cam [46] and Nafees [47] et al. identified citric acid as the predominant organic acid in pomegranates. These findings align with emerging models of fruit flavor determination that emphasize genetic control of acid metabolism as the primary driver of flavor diversification among varieties [48,49].

The composition of soluble sugars and organic acids plays a pivotal role in determining pomegranate fruit flavor profiles [42]. Considering the inherent complexity of metabolic networks that govern these compounds and the limited impact of individual gene expression on overall metabolite accumulation [50], our investigation adopted an integrated systems biology approach to elucidate key regulatory components. Through comprehensive network analysis, we successfully identified and characterized three candidate genes implicated in soluble sugar metabolic pathways and eight genes involved in the regulation of organic acid metabolism.

Our analysis identified *FBA1*, *SS*, and *SWEET* genes predominantly involved in sucrose metabolism and soluble sugar transport processes governing pomegranate sugar accumulation. Fructose-bisphosphate aldolase (FBA) catalyzes the cleavage of fructose-1,6-bisphosphate in glycolysis and gluconeogenesis [51]. In watermelon, studies have identified *FBA2* as a key regulator of sugar metabolism through multi-omics approaches [52]. Consistent with reports in ‘TSH’ and ‘MD’ pomegranate [42], *FBA1* expression progressively declined during fruit development in both varieties in this study. This conserved downregulation pattern suggests a metabolic shift where reduced *FBA1*-mediated glycolytic activity redirects carbon flux toward hexose accumulation, reinforcing its role in sugar metabolism regulation.

Sucrose synthase (SS), which catalyzes the reversible cleavage of sucrose into fructose and UDP-glucose [53], has been widely reported to exhibit a positive correlation between its gene expression and sucrose accumulation in tomato [54] and blueberry [55], suggesting its predominant role in sucrose synthesis. Strikingly, our findings reveal an inverse regulatory pattern in pomegranate, where the expression levels of *SS* genes exhibited a significant negative correlation with sucrose content (r = −0.62, *p* = 0.005). This divergence implies a potential evolutionary shift in *SS* functionality, wherein *SS* may preferentially drive sucrose degradation [55,56].

Beyond metabolic enzymes, sugar transporters, particularly SWEET family proteins, play pivotal roles in regulating soluble sugar accumulation by mediating transmembrane translocation of soluble sugars from phloem to flesh cells during fruit development [12,57]. Overexpression of *SWEET7* in stable transgenic tomato lines significantly increased total sugar content [58]. In peach, *PpSWEET9a* and *PpSWEET14* mediated sucrose efflux, driving sucrose allocation from source leaves to fruits and promoting sugar accumulation [59]. In this study, *SWEET16* exhibited significant upregulation across developmental stages in both varieties, demonstrating strong positive correlations with glucose and fructose accumulation. These findings substantiate its crucial function in both phloem unloading and vacuolar sequestration of sugars [60,61].

The metabolic regulation of organic acids involves complex compartmentalization across cytoplasm, mitochondria, and glyoxysomes for synthesis and degradation before vacuolar storage [62,63]. Although citric acid homeostasis is conventionally maintained through a balance between synthesis and catabolism, our transcriptomic analysis specifically identified differentially expressed genes (*ADH1*, *GABP*, *ICL* and *PDC4*) associated with citric acid catabolic pathways. This finding suggests that variety-dependent variation in citric acid accumulation in pomegranate is principally regulated at the level of catabolic activity rather than biosynthetic capacity.

Alcohol dehydrogenase (ADH) catalyzes the reduction of acetaldehyde to ethanol, diverting pyruvate from mitochondrial acetyl-CoA synthesis and limiting substrate availability for citric acid production. In litchi, coordinated expression of *ADH1* and citric acid accumulation during early fruit development suggests its potential role in citric acid biosynthesis [64]. In contrast, our study reveals a functionally distinct regulatory mechanism in pomegranate. The transcription abundance of the *ADH1* structural gene in ‘Azadi’ was strikingly higher than that in ‘Sharp Velvet’ at 90 and 135 DAF. The low expression of *ADH1* gene may promote cytoplasmic pyruvate accumulation through negative feedback regulation, with subsequent mitochondrial translocation and conversion to acetyl-CoA fueling citric acid synthesis [65], suggesting that *ADH1* may indirectly regulate citric acid homeostasis through its influence on TCA cycle substrate availability, mirroring the well-documented crosstalk between fermentative metabolism and citric acid accumulation previously reported in peach [66].

The GABA shunt pathway plays a crucial role in organic acid metabolism by facilitating the conversion of cytoplasmic citric acid to succinate through mitochondrial GABA transport mediated by GABP proteins [67,68,69]. Although this pathway has been well documented in citrus [70,71] and peach [66], where *GABP* upregulation enhances citric acid accumulation through GABA-driven replenishment of tricarboxylic acid cycle (TCA cycle) intermediates [67], our study uncovered a functionally divergent mechanism in pomegranate. Here, we identify the *GABP* gene in pomegranate, with all four homologs exhibiting strong negative correlations with citric acid content. There is no report regarding this negative relationship between GABP expression and citric acid content. Therefore, we propose that high *GABP* expression accelerates mitochondrial GABA influx to promote citric acid degradation. The characterization of *GABP* genes in pomegranate needs future investigation of GABP protein–protein interactions through yeast two-hybrid screening or co-immunoprecipitation to elucidate its regulatory network.

Isocitric acid lyase (ICL) plays a dual metabolic role by catalyzing the cleavage of isocitric acid into succinate and glyoxylate in the glyoxylate cycle while simultaneously generating malic acid for gluconeogenesis [16]. Interestingly, our data revealed higher *ICL* expression levels in ‘Sharp Velvet’ compared to ‘Azadi’ across three developmental stages, despite significantly elevated citric acid accumulation in ‘Sharp Velvet’. This apparent paradox, where the high acid content variety maintained both elevated *ICL* expression and citric acid levels through maturation, suggests potential post-translational modification of ICL enzyme activity. Collectively, these findings strongly suggest that citric acid degradation in pomegranate occurs predominantly through the GABA shunt pathway.

The pyruvate dehydrogenase complex (PDC) serves as a crucial metabolic node by catalyzing the irreversible conversion of pyruvate to acetyl-CoA, with its activity being negatively regulated through phosphorylation by pyruvate dehydrogenase kinase (PDK). Notably, in peaches, PDK overexpression within acidity-related QTL intervals promoted citric acid accumulation, suggesting a regulatory link between PDC suppression and organic acid metabolism [66,72]. In our study, *PDC4* exhibited distinct variety-specific expression patterns, with significantly higher transcript levels in ‘Azadi’ during ripening stages. This elevated expression likely enhances pyruvate catabolism, thereby reducing the substrate pool available for citric acid biosynthesis. While vacuolar proton pumps critically establish the electrochemical gradient for citric acid sequestration [63,73], the absence of differentially expressed transporters in our data implied that citric acid accumulation was primarily governed by metabolic flux regulation rather than vacuolar storage capacity [16,66,74].

The *ME1* gene associated with malic acid metabolism was also identified. Malic enzyme (ME) plays a pivotal role in organic acid metabolism by catalyzing the oxidative decarboxylation of malic acid to pyruvate [16]. In plants, this reaction is mediated by two distinct cofactor-specific isoforms: NAD-dependent ME (NAD-ME) and NADP-dependent ME (NADP-ME), which differentially regulate energy metabolism and carbon flux partitioning [75]. *ME* gene expression has been reported to exhibit a significant negative correlation with malic acid content in plum [75] and litchi [37], implying a potential functional convergence in malic acid metabolism between these two species. However, our study uncovers a divergent regulatory paradigm in pomegranate. Notably, at 120 DAF, both malic acid content and *ME1* expression in the variety ‘Sharp Velvet’ were significantly lower than those in ‘Azadi’. This unexpected co-decline implies that malic acid metabolism in pomegranate may be governed by multi-gene coordination.

Recent studies have established the critical roles of bHLH, ERF and WRKY TFs in regulating organic acid metabolism [76,77,78]. Notably, although the *CitbHLH2*-*CitMYB52* complex negatively regulated citric acid accumulation by suppressing *CitALMT* in citrus [63], the identified *bHLH42* in pomegranate displayed positive regulatory effects (r = 0.78, *p* = 0.001), suggesting bHLH members may employ distinct protein complexes for pathway-specific modulation. Similarly, *ERF4* and *ERF062* exhibited opposing effects on citric acid accumulation, mirroring observations in ponkan where *CitERF13* promotes while *CitERF6* inhibits citric acid accumulation [79], potentially through differential regulation of vacuolar proton pumps or metabolic enzymes. Our results demonstrated that *WRKY6* and *WRKY23* exhibited significant negative correlations with citric acid content, while *WRKY28* showed a positive correlation (r = 0.68, *p* = 0.002), consistent with previous reports in citrus [80] and pear [81], indicating functional divergence among WRKY family members in citric acid regulation [32]. These findings reveal specialized transcriptional networks controlling citric acid homeostasis. Practically, the citric acid-associated transcription factors enable marker-assisted selection (MAS) to breed low-acid varieties by targeting their expression patterns.

While this study provides novel insights into the molecular regulation of sugar and acid metabolism in pomegranate, several limitations should be acknowledged. First, although candidate genes and transcription factors were identified, functional validation through genetic manipulation remains to be performed to confirm their roles in metabolite accumulation. Second, while transcriptomic and physiological data were integrated, complementary metabolomic or proteomic analyses could further resolve post-transcriptional regulatory layers. Addressing these limitations in future studies will strengthen the mechanistic understanding of flavor compound regulation in pomegranates.

## 5. Conclusions

In summary, this study employed comparative transcriptome analysis between the pomegranate variety ‘Sharp Velvet’ with high acid content and ‘Azadi’ with low acid content to elucidate key genes governing soluble sugars and organic acids metabolism. Our findings demonstrated that flavor divergence between these varieties primarily arose from differential citric acid accumulation. Citric acid dynamics were driven by metabolic flux regulation during synthesis and degradation via the GABA shunt pathway. Transcriptomic profiling revealed 11 candidate genes in soluble sugars and organic acid metabolism; further, WGCNA specifically pinpointed six TFs regulating citric acid accumulation. Future investigations should prioritize the functional characterization of these candidates and their interaction networks to comprehensively understand soluble sugars and organic acid regulation in pomegranate.

## Figures and Tables

**Figure 1 foods-14-01755-f001:**
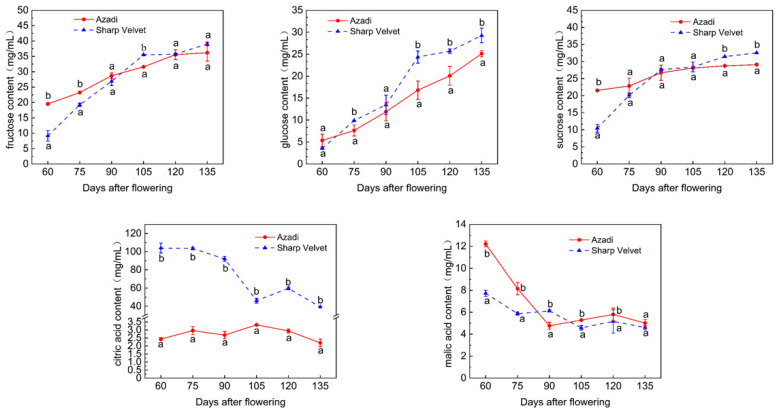
Change of fructose, sucrose, glucose, citric acid and malic acid content in two varieties with development. Different lowercase letters indicate significant differences between varieties in the same developmental stages (*p* < 0.05, *n* = 3).

**Figure 2 foods-14-01755-f002:**
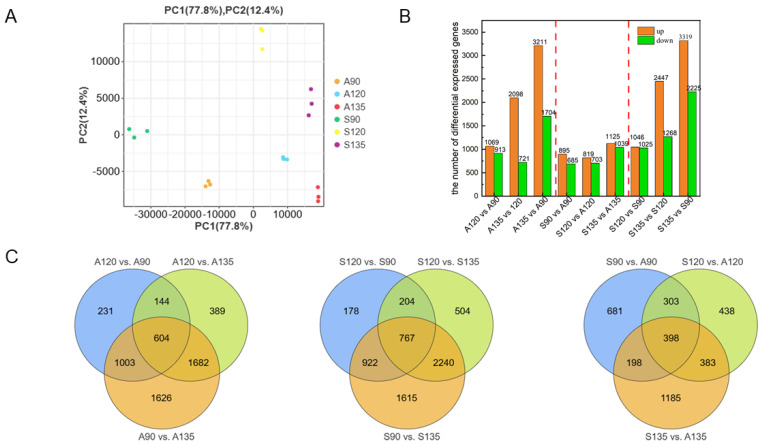
PCA analysis of samples and DEGs in comparison groups. (**A**). PCA analysis of samples. (**B**). The number of DEGs in different periods of the same variety and different varieties in the same period. (**C**). The DEGs were identified in pairwise comparisons in different periods of the same variety and different varieties in the same period.

**Figure 3 foods-14-01755-f003:**
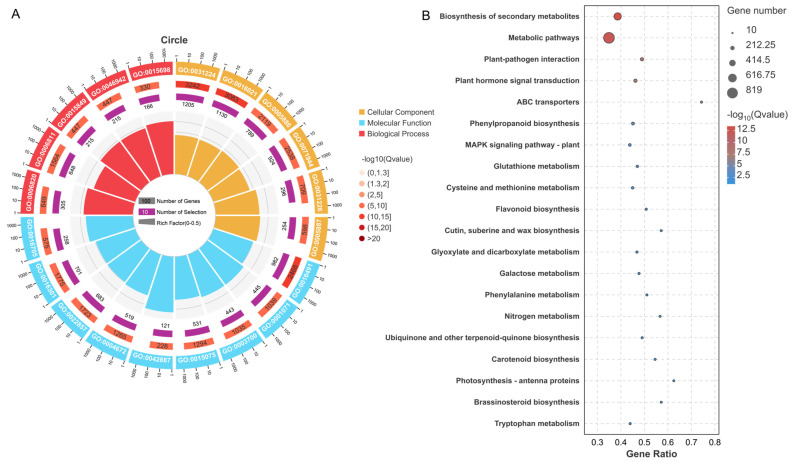
Enrichment analysis of DEGs in all comparison groups. (**A**). GO functional enrichment analysis for the DEGs in all comparison groups. (**B**). The statistics of KEGG enrichment of the DEGs in all comparison groups.

**Figure 4 foods-14-01755-f004:**
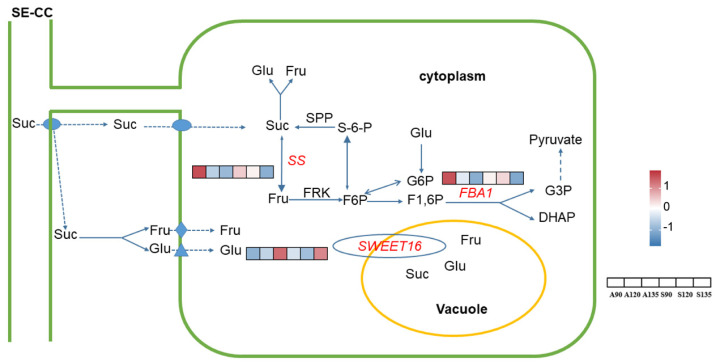
The expression profile of candidate genes involved in soluble sugar metabolic pathway in pomegranate. The heatmap of each DEG was constructed based on log_2_ (TPM) and the values were rowscaled. The color range from red to blue represents the log_2_ (TPM) value, in which red color denotes genes with high expression levels, and blue color means low expression. DHAP, dihydroxyacetone phosphate; Fru, fructose; F6P, fructose–6–phosphate; F1,6P, fructose–1,6–phosphate; Glu, glucose; G3P, glyceraldehyde 3–phosphate; G–6–P, glucose–6–phosphate; Suc, sucrose; S–6–P, sucrose–6–phosphate;SPP, sucrose phosphatase phosphorylase. The probable direction of reversible reactions is indicated by the large arrow.

**Figure 5 foods-14-01755-f005:**
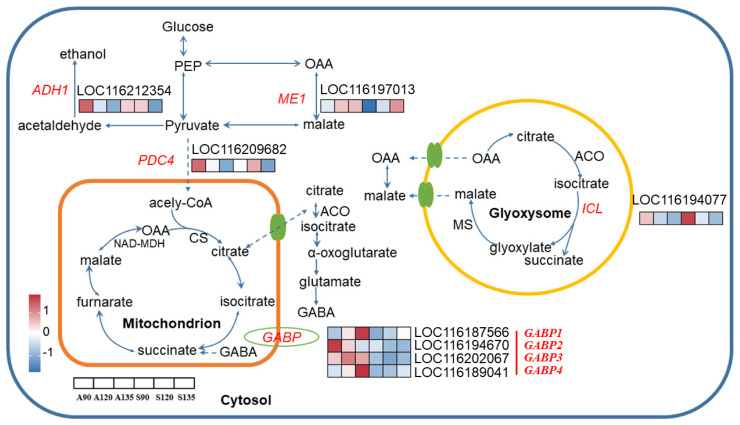
The expression profile of candidate genes involved in the organic acid metabolic pathway in pomegranate. The heatmap of each DEG was constructed based on log_2_ (TPM) and the values were rowscaled. The color range from red to blue represents the log_2_ (TPM) value, in which red color denotes genes with high expression levels, and blue color means low expression. ACO, aconitase; ADH, alcohol dehydrogenase; CS, citric acid synthase; GABP, γ–aminobutyric acid transport; ICL, isocitric acid lyase; ME, malic acid enzyme; OAA, oxaloacetate; PDC, pyruvate Decarboxylase; PEP, phosphoenolpyruvate. The probable direction of reversible reactions is indicated by the large arrow.

**Figure 6 foods-14-01755-f006:**
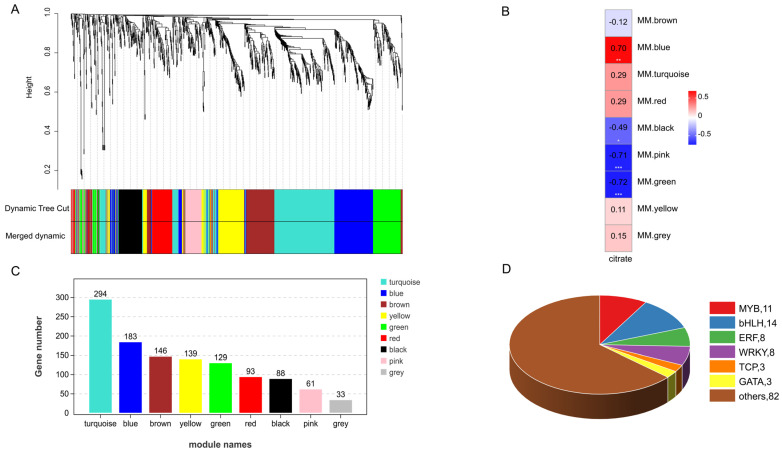
WGCNA identification of the TFs involved in citric acid regulation in ‘Azadi’ and ‘Sharp Velvet’ pomegranate. (**A**) The cluster dendrograms of co–expressed gene modules. (**B**) Correlation heatmap between module and citric acid content. * indicates *p* < 0.05, ** indicates *p* < 0.01, *** indicates *p* < 0.001. (**C**) Number of genes in each module. (**D**) Number of TF family genes in green module.

**Figure 7 foods-14-01755-f007:**
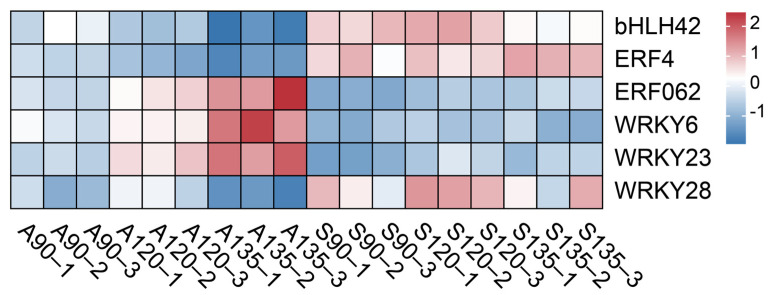
Expression profile of six candidate TFs related to citric acid biosynthesis.

**Figure 8 foods-14-01755-f008:**
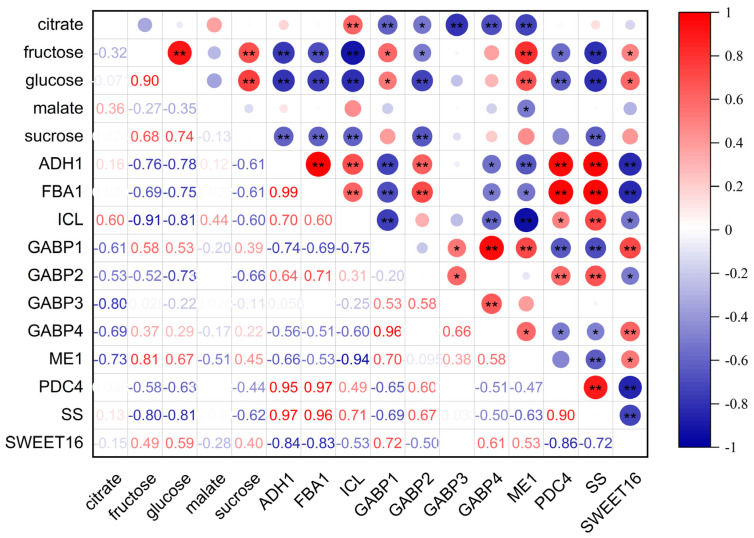
Correlation analysis between gene expression levels and the content of soluble sugars and organic acids. Circle size represents significance level. Red and blue colors denote positive and negative correlations, respectively. * indicates *p* < 0.05, ** indicates *p* < 0.01.

**Figure 9 foods-14-01755-f009:**
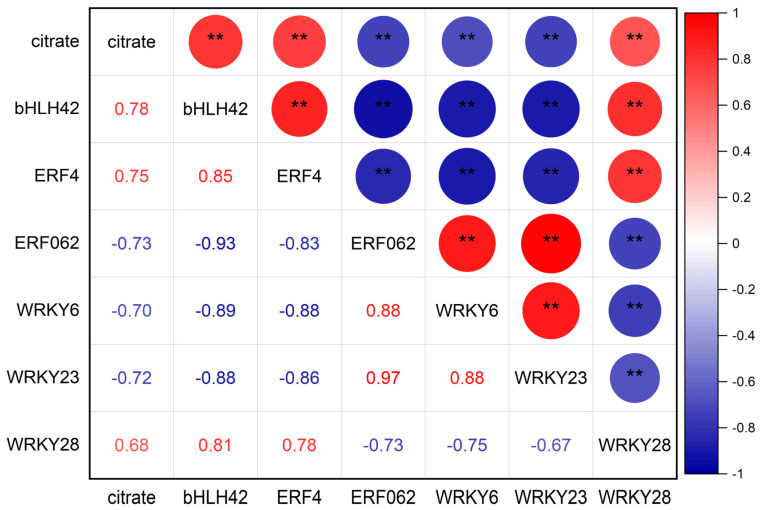
Correlation analysis between TFs expression levels and citric acid content. Circle size represents significance level. Red and blue colors denote positive and negative correlations, respectively. * indicates *p* < 0.05, ** indicates *p* < 0.01.

**Figure 10 foods-14-01755-f010:**
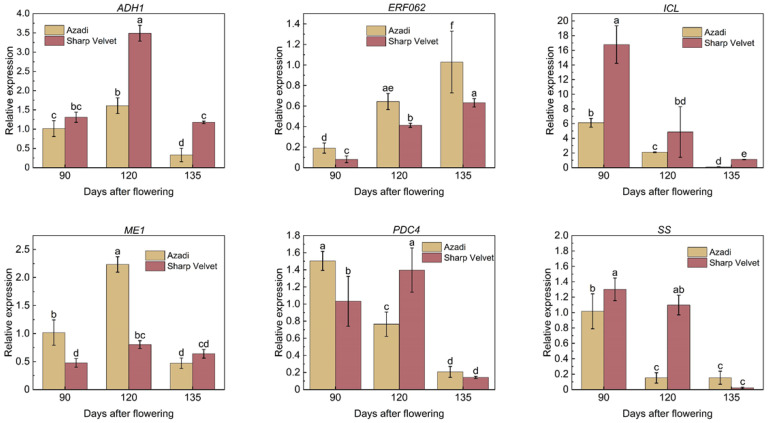
Transcriptional levels of candidate genes by qRT-PCR. The *x*-axis in each chart stands for the three developmental stages (90, 120, 135 DAF) of two varieties. Bars with different letters indicate significant differences at *p* < 0.05 level according to Duncan’s test.

## Data Availability

The RNA-seq data in this study have been deposited into the NCBI Sequence Read Archive (SRA) database under the BioProject with accession number PRJNA1198205 for ‘Azadi’ and ‘Sharp velvet’.

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
