# Peer review of "Transcriptomic Profiling Uncovers Molecular Basis for Sugar and Acid Metabolism in Two Pomegranate (Punica granatum) Varieties"

_foods, 2025, doi:10.3390/foods14101755_

Round 1
Reviewer 1 Report
Comments and Suggestions for Authors
This study analyzed transcriptomes of two pomegranate cultivars, identifying key genes and transcription factors involved in citrate metabolism. The GABA shunt pathway and six transcription factors were highlighted as critical regulators, offering a molecular basis for breeding. The manuscript requires major revisions to enhance clarity and scientific sound.
The abstract includes general background information that is more appropriate in the introduction. It is recommended for authors to condense this section and focus more on the research findings and mechanisms in the abstract.
Line 83, additional explanation is needed regarding the representativeness of the two selected cultivars, ‘Sharp Velvet’ (high acid) and ‘Azadi’ (low acid).
Line 93, the authors must provide a more detailed description of the statistical analysis used.
Line 176, in "29.08 mg/ml and 32.53 mg/ml", the unit "mg/ml" is repeated and can be omitted after the first value.
Line 280, please improve the resolution of Fig. 8.
Line 307, the authors should more strongly emphasize the novelty and contribution of this study compared to previous research.
Line 329, the authors are recommended to include the p-values corresponding to the correlation results.
Author Response
Dear Reviewer,
We appreciate the opportunity to revise our manuscript titled " Transcriptomic Profiling Uncovers Molecular Basis for Sugar and Acid Metabolism in Two Pomegranate (Punica granatum) Varieties." and are grateful for the insightful comments provided. Those comments are all valuable and very helpful for revising and improving our paper, as well as the important guiding significance to our researches. In the following, we have provided detailed responses to each of the comments. Revised portion are marked in yellow in the paper. We have tried our best to make all the revisions clear, and we hope that the revised manuscript meets the requirements for publication.
Comments 1: The abstract includes general background information that is more appropriate in the introduction. It is recommended for authors to condense this section and focus more on the research findings and mechanisms in the abstract.
Response 1: We sincerely thank the reviewer for this constructive suggestion. We have comprehensively revised the abstract to focus on core findings in Page 1, line 13-25.
Comments 2: Line 83, additional explanation is needed regarding the representativeness of the two selected cultivars, ‘Sharp Velvet’ (high acid) and ‘Azadi’ (low acid).
Response 2: We have added explanations about the representativeness of the selected cultivars. The detailed description was in Page 3, line 101-103.
Comments 3: Line 93, the authors must provide a more detailed description of the statistical analysis used.
Response 3: We have now added a detailed description in the Materials and Methods section in Page 4, Lines 178-183.
Comments 4: Line 176, in "29.08 mg/ml and 32.53 mg/ml", the unit "mg/ml" is repeated and can be omitted after the first value.
Response 4: As suggested, we have revised the text to eliminate the repeated unit "mg/ml" in Page5, Line 191-192.
Comments 5: Line 280, please improve the resolution of Fig. 8.
Response 5: In response to the concern regarding figure resolution, all figures, including Figure 8, have been re-exported as high-resolution (600 dpi) with enlarged labels, optimized color schemes, and enhanced contrast to ensure clarity in Page 9, Lines 313.
Comments 6: Line 307, the authors should more strongly emphasize the novelty and contribution of this study compared to previous research.
Response 6: In the revised manuscript, we have expanded the discussion around Line 307 to explicitly highlight the unique aspects of our work compared to previous studies in Page 12, Lines 345.
Comments 7: Line 329, the authors are recommended to include the p-values corresponding to the correlation results.
Response 7: We have now added the corresponding p-values for all correlation results to provide full statistical transparency.
We hope that the changes we've made resolve all your concerns about the article. Thank you for your positive comments and valuable suggestions to improve the quality of our manuscript.
Thank you again and best regards!
Yours sincerely,
Xueqing Zhao
Reviewer 2 Report
Comments and Suggestions for Authors
The introduction is solid, well-founded, with an excellent bibliographic basis and logical progression. Small adjustments in the depth of the specific references to pomegranate and in the link with practical application would give it the finishing touch.
The experimental design is adequate, using two pomegranate cultivars with different acidity levels and sampling at fruit development stages. The combination of analyses is effective, but could be strengthened with information on environmental control and the total number of fruits collected per stage. The methods are well described, but it would be useful to include details on RNA quality control and the software used in the statistical analyses, increasing transparency and reproducibility.
The presentation of the results is detailed, but can be considered a little complex due to the large volume of data and the technical terminology used. The structure of the text is well organized, but the density of information can make it challenging to read for those unfamiliar with the subject.
The conclusions are well supported by the results, although there could be more specific details about the relationship between the identified genes and the biological mechanisms, which would help to further strengthen the correlation between results and conclusions.
The research is relevant and addresses an interesting area, although it is not entirely innovative in the field of transcriptomic analysis. The content is of great importance, as it may impact the genetic improvement of pomegranate cultivars. The presentation is clear, but could be more detailed, especially in the methods and data analysis. The scientific soundness is good, with the use of recognized techniques, but could be reinforced with more evidence. The study is of interest mainly to researchers in the field of biotechnology, with limited potential impact outside this niche. The overall merit is relevant, but could be improved with more experimental data and a deeper discussion of the practical implications.
Most of the references are recent, with emphasis on studies published between 2021 and 2025, followed by a considerable number of articles from 2011 to 2020. There are also some citations of older studies, but most of the sources are current, indicating very up-to-date research.
The file has been subjected to an anti-plagiarism program, the results of which are attached.

Author Response
Dear Reviewer,
We appreciate the opportunity to revise our manuscript titled " Transcriptomic Profiling Uncovers Molecular Basis for Sugar and Acid Metabolism in Two Pomegranate (Punica granatum) Varieties." and are grateful for the insightful comments provided. Those comments are all valuable and very helpful for revising and improving our paper, as well as the important guiding significance to our researches. In the following, we have provided detailed responses to each of the comments. Revised portion are marked in emerald green in the paper. We have tried our best to make all the revisions clear, and we hope that the revised manuscript meets the requirements for publication.
Comments 1: The introduction is solid, well-founded, with an excellent bibliographic basis and logical progression. Small adjustments in the depth of the specific references to pomegranate and in the link with practical application would give it the finishing touch.
Response 1: Thank you for your insightful feedback. We have revised the Introduction to enhance specificity regarding pomegranate-focused studies and strengthened connections to practical applications in Page 2, line 81-86.
Comments 2: The experimental design is adequate, using two pomegranate cultivars with different acidity levels and sampling at fruit development stages. The combination of analyses is effective, but could be strengthened with information on environmental control and the total number of fruits collected per stage. The methods are well described, but it would be useful to include details on RNA quality control and the software used in the statistical analyses, increasing transparency and reproducibility.
Response 2: In the revised manuscript, we have enhanced the materials and methods section by adding details on environmental controls during sample collection, specifying the total number of fruits collected at each stage, supplementing RNA quality control protocols, and providing software information used in data analysis. These revisions aim to improve the transparency and reproducibility of the study in Page 3, line 103-104 ,166-168 and 178-183.
Comments 3: The presentation of the results is detailed, but can be considered a little complex due to the large volume of data and the technical terminology used. The structure of the text is well organized, but the density of information can make it challenging to read for those unfamiliar with the subject.
Response 3: The revised manuscript introduces transcriptomics in the Introduction section, explaining its definition and applications to ensure readers understand this technology in Page 2, line 67-76.
Comments 4: The conclusions are well supported by the results, although there could be more specific details about the relationship between the identified genes and the biological mechanisms, which would help to further strengthen the correlation between results and conclusions.
Response 4: In this study, we analyzed the regulatory processes in which the key screened genes are involved. Combined with their transcriptional expression levels, we hypothesized functional differences and potential regulatory divergences of these genes between the two varieties. These findings were further discussed in the discussion section in Page 12, line 345.
Comments 5: The research is relevant and addresses an interesting area, although it is not entirely innovative in the field of transcriptomic analysis. The content is of great importance, as it may impact the genetic improvement of pomegranate cultivars. The presentation is clear, but could be more detailed, especially in the methods and data analysis. The scientific soundness is good, with the use of recognized techniques, but could be reinforced with more evidence. The study is of interest mainly to researchers in the field of biotechnology, with limited potential impact outside this niche. The overall merit is relevant, but could be improved with more experimental data and a deeper discussion of the practical implications.
Response 5: We supplemented the methodology and data analysis in Page 4, line 178-183., and refined the discussion section in Page 12, line 345, with the aim of interpreting and exploring the potential regulatory mechanisms underlying the findings of this study . This work thereby lays the foundation for further investigation into the regulatory mechanisms of sugar-acid metabolism.
Comments 6: Most of the references are recent, with emphasis on studies published between 2021 and 2025, followed by a considerable number of articles from 2011 to 2020. There are also some citations of older studies, but most of the sources are current, indicating very up-to-date research.
Response 6: To track the latest research advances in sugar-acid metabolism mechanisms, this study incorporates references to literature published within the last five years, along with seminal works in the field. Furthermore, additional relevant literature has been added to the revised manuscript in response to reviewers' comments.
We hope that the changes we've made resolve all your concerns about the article. Thank you for your positive comments and valuable suggestions to improve the quality of our manuscript.
Thank you again and best regards!
Yours sincerely,
Xueqing Zhao
Reviewer 3 Report
Comments and Suggestions for Authors
The authors integrated physiological and transcriptomic analyses of two pomegranate cultivars to characterize changes in soluble sugars and organic acids during fruit development. It identified citrate accumulation via the GABA shunt as key to acidity differences and uncovered 11 candidate metabolic genes and six core transcription factors. These findings offer new insights into fruit acidity regulation and molecular targets for breeding.
The paper is good, but some changes will be helpful for improving its quality.
Line 139. Citation is out of order. It should be [35], not [74]. Please fix it.
A “Data Analysis section should be included in the Materials and Methods.
I also encourage to increase the number of citation in the Methods Section. There are entire paragraphs without citations.
What do the letters in Figures 1, 10 stand for? Is there any statistical significance?
Figure 1. Please add “n” for each group. This, for all samples, should appear in a table that could be located in Supplementary Materials.
Figures 3,6 needs better resolution.
Figure 4 requires more explanation. It is not absolutely clear how it works. What is the scale?
Figure 8. Please indicate what represents the size of the circle? The scale for correlation should be labeled. Is the correlation parametric? No parametric?
Figure 9. Use “p” instead of “P”.
Line 119. Please separate the number as this one: “(0.1, 0.2, 0.4, 0.8 mg/mL)”.
In the Discussion Section please focus on trying to explain why the results matter, avoid repeating what was found.
Line 146. “The online website (https://www.omicsmart.com) was applied to PCA 146 and heat maps analysis”. Are you sure? Please rephrase.
Line 198. Remove "of" after "comparisons".
A paragraph on Limitations should be good.
The conclusions should be more concise, and there should be a section on limitations of the study.
Author Response
Dear Reviewer,
We appreciate the opportunity to revise our manuscript titled " Transcriptomic Profiling Uncovers Molecular Basis for Sugar and Acid Metabolism in Two Pomegranate (Punica granatum) Varieties." and are grateful for the insightful comments provided. Those comments are all valuable and very helpful for revising and improving our paper, as well as the important guiding significance to our researches. In the following, we have provided detailed responses to each of the comments. Revised portion are marked in blue in the paper. We have tried our best to make all the revisions clear, and we hope that the revised manuscript meets the requirements for publication.
Comments 1: Line 139. Citation is out of order. It should be [35], not [74]. Please fix it.
Response 1: We sincerely thank the reviewer for catching this citation error. In the revised manuscript, the misplaced reference has been corrected to accurately reflect the intended citation. All subsequent references have been renumbered accordingly to maintain sequential order. This revision ensures full consistency with the bibliography in Page 4, line 144.
Comments 2: A “Data Analysis section should be included in the Materials and Methods.
Response 2: The revised manuscript has been expanded in the Materials and Methods section to include a dedicated data analysis subsection in Page 4, line 178-183.
Comments 3: I also encourage to increase the number of citation in the Methods Section. There are entire paragraphs without citations.
Response 3: The revised manuscript has been updated in the Methods section with the addition of citations to enhance the scientific rigor and reproducibility of the experiments in Page 3, line 115 and Page 4, line 159,179-183.
Comments 4: What do the letters in Figures 1, 10 stand for? Is there any statistical significance?
Response 4: In Figures 1 and 10, we elaborate on the interpretation of numerical labels and the statistical methods employed in Page 5, line 198-199 and Page 12 line 343-344.
Comments 5: Figure 1. Please add “n” for each group. This, for all samples, should appear in a table that could be located in Supplementary Materials.
Response 5: We have added the sample sizes per group in Figure 1 and supplemented the raw data of sugar and acid components in Supplementary Table S2 in Page 5, line 199.
Comments 6: Figures 3,6 needs better resolution.
Response 6: In response to the concern regarding figure resolution, all figures, including Figure 3,6 have been re-exported as high-resolution (600 dpi) with enlarged labels, optimized color schemes, and enhanced contrast to ensure clarity in Page 7, line 250 and Page 9, line 309.
Comments 7: Figure 4 requires more explanation. It is not absolutely clear how it works. What is the scale?
Response 7: Figure 4,5 has been revised with enhanced annotations, a refined depiction of the sugar metabolic pathway, and the addition of a scale bar explanation to improve interpretability in Page 8, line 277-281, 284-288.
Comments 8: Figure 8. Please indicate what represents the size of the circle? The scale for correlation should be labeled. Is the correlation parametric? No parametric?
Response 8: As suggested, Figure 8 has been revised to include explanations of the circle symbols, additional correlation parameters, and a scale bar with unit annotations to enhance data clarity in Page 10, line 325-326 and Page 11, line 334-335.
Comments 9: Figure 9. Use “p” instead of “P”.
Response 9: As suggested, we have revised the text to use ‘p’ instead of ‘P’ in Page 11, line 335.
Comments 10: Line 119. Please separate the number as this one: “(0.1, 0.2, 0.4, 0.8 mg/mL)”.
Response 10: We have revised the concentration values to separate numbers with commas and spaces as suggested in Page 3, line 126.
Comments 11: In the Discussion Section please focus on trying to explain why the results matter, avoid repeating what was found.
Response 11: In the revised manuscript, we have refined the Discussion section to emphasize the significance of the results while avoiding redundancy with findings presented earlier in Page 12, line 345. These revisions enhance the readability of the text and better highlight the novel contributions of the study.
Comments 12: Line 146. “The online website (https://www.omicsmart.com) was applied to PCA 146 and heat maps analysis”. Are you sure? Please rephrase.
Response 12: We have carefully reviewed the web-based tools used for generating the heatmap and PCA plots, and have implemented necessary corrections in the revised manuscript to ensure the accuracy and reproducibility of these visualizations in Page 4, line 153.
Comments 13: Line 198. Remove "of" after "comparisons".
Response 13: We have revised by removing the preposition "of" after "comparisons" to correct the grammatical structure in Page 6, line 223.
Comments 14: A paragraph on Limitations should be good.
Response 14: As suggested, we have revised the text to add Limitations section in Page 15, line 480-487.
Comments 15: The conclusions should be more concise, and there should be a section on limitations of the study.
Response 15: As suggested, we have incorporated a discussion of study limitations into the Conclusion section of the revised manuscript in Page 15, line 491-494, 497-499.
We hope that the changes we've made resolve all your concerns about the article. Thank you for your positive comments and valuable suggestions to improve the quality of our manuscript.
Thank you again and best regards!
Yours sincerely,
Xueqing Zhao
Round 2
Reviewer 1 Report
Comments and Suggestions for Authors
All comments have been appropriately addressed.